# Is the apparently protective effect of maternal nicotine replacement therapy (NRT) used in pregnancy on infant development explained by smoking cessation?: secondary analyses of a randomised controlled trial

Barbara Iyen,[1] Luis R Vaz,[1] Jaspal Taggar,[1] Sue Cooper,[1] Sarah Lewis,[2] Tim Coleman[1]

[1]Division of Primary Care, University of Nottingham, Nottingham, UK
[2]Division of Epidemiology and Public Health, University of Nottingham, Nottingham, UK

**Correspondence to**
Dr Barbara Iyen;
barbara.iyen@nottingham.ac.uk

## ABSTRACT

**Objective** To investigate relationships between maternal smoking status in pregnancy and infant development. The largest randomised controlled trial of nicotine replacement therapy (NRT) for smoking cessation in pregnancy, the smoking, nicotine and pregnancy (SNAP) trial, found that at 1 month after randomisation, smoking cessation rates were doubled in the NRT group compared with the placebo group. At delivery, there was no significant difference in cessation rates between groups. Surprisingly, infants born to women randomised to NRT were more likely to have unimpaired development at 2 years. We hypothesised that this apparently protective effect was due to smoking cessation caused by NRT and so, investigate this relationship using the same cohort.

**Design** Secondary analysis of a randomised controlled trial.

**Setting** Seven antenatal hospitals in the Midlands and North-West England.

**Participants** Eight hundred and eighty-four pregnant smokers randomised to receive either NRT patches or visually-identical placebo in the SNAP trial. Participants' smoking behaviour were recorded at randomisation, 1 month after their target quit date and at delivery.

**Methods** Using logistic regression models, we investigated associations between participants' smoking measures and infant development (assessed using the Ages and Stages questionnaire) at 2 years.

**Main outcome measures** 2 year infant development.

**Results** Developmental impairment was reported for 12.7% of study 2 year olds. Maternal heaviness of smoking at randomisation (OR: 1.26, 95% CI: 0.82 to 1.96, p=0.091), validated smoking abstinence recorded at 1 month after a quit date (OR: 1.02, 95% CI: 0.60 to 1.74, p=0.914) and validated smoking abstinence recorded at both 1 month after a quit date and at the end of pregnancy (OR: 1.52, 95% CI: 0.81 to 2.85, p=0.795) were not independently associated with infant developmental impairment at 2 years.

**Conclusion** We found no evidence that NRT treatment improved infants' developmental outcomes through smoking cessation.

**Trial registration number** CTA03057/0002/001-0001; Post-results

## Strengths and limitations of this study

► This study uses data from the smoking, nicotine and pregnancy (SNAP) trial, the largest randomised controlled trial of nicotine replacement therapy for smoking cessation in pregnancy.

► The SNAP trial remains the only maternal smoking cessation trial to investigate infant outcomes beyond delivery.

► Baseline characteristics and potential confounders were included in multivariable analyses to determine independent associations between maternal smoking and 2 year developmental outcomes.

► Maternal factors which were not measured in the SNAP trial could not be adjusted for in multivariable analyses.

## INTRODUCTION

Smoking in pregnancy is associated with increased risks of many adverse outcomes including miscarriage, stillbirth, prematurity, low birth weight, perinatal morbidity and mortality[1] and is a substantial international public health problem; in high income countries 13% to 25% of pregnant women smoke[2–5] and in developing countries,[6 7] the WHO predicts a future epidemic. Fortunately, impacts on the foetus are avoidable and cessation in pregnancy improves infants' birth weights and reduces the risk of premature birth.[8] Additionally, pregnancy is the life event which seems to most motivate smokers' cessation attempts; over 50% of UK pregnant smokers try stopping.[5] Women are, therefore, likely to be receptive to support with stopping and, as it is effective for non-pregnant smokers[9] and possibly also for pregnant smokers too, nicotine replacement therapy

(NRT) is frequently offered to pregnant women who smoke.[10] Although NRT is not thought to be completely risk-free because it exposes users to nicotine, this is considered safer than smoking because smokers are already nicotine-exposed and NRT users are not exposed to the numerous harmful chemicals and tobacco smoke carcinogens such as carbon monoxide, tar and lead.

Animal experiments and human-subject laboratory investigations suggest that nicotine may adversely affect the foetal cardiovascular system and developing nerve tissue,[11 12] however trials of NRT for smoking cessation in pregnancy have been permitted because any foetal harm from the nicotine would likely be compensated for by the benefits of mothers stopping smoking.[13] In the largest of these trials, compliance rates in both the NRT and the placebo arms were low, with only 7.2% of women assigned to receive NRT and 2.8% of women assigned to receive placebo, using patches for more than 1 month. Nicotine patches doubled cessation rates at 4 weeks but by delivery there was no significant difference in cessation rates between the NRT and placebo groups.[14] Unexpectedly, infants born to NRT group women were 40% more likely to have unimpaired development at 2 years of age than infants born to women in the placebo groups[15] (OR: 1.40, 95% CI: 1.05 to 1.86, p=0.023) and there was a dose-response relationship between adherence to NRT and impairment-free infant development.[15] It seems implausible that this effect would be due to nicotine having a direct beneficial effect on the foetus, and there is a possibility that the observed effect might be due to differences in unmeasured characteristics between the participants in the NRT group and those in the placebo group. However, as NRT had a substantial impact on cessation in early pregnancy, we hypothesise that the better developmental outcomes in NRT group infants and the dose-response relationship between increasing NRT use and these might be explained by smoking cessation during pregnancy caused by NRT use. Consequently, we present secondary analyses of data from the same trial to investigate whether the absence of infants' developmental impairments at 2 years was associated with maternal smoking status measured at different points in the trial.

## METHODS
### Study participants
We used data from the smoking, nicotine and pregnancy (SNAP) trial, a placebo randomised-controlled trial of NRT for smoking cessation in pregnant women.[14] The trial recruited 1050 participants aged 16 to 45 years between 12 and 24 weeks gestation who smoked at least 10 cigarettes a day before pregnancy and at least five cigarettes a day during pregnancy, with exhaled carbon monoxide (CO) readings of ≥8 ppm. Participants were randomised to receive up to an 8 week course of either 15 mg per 16 hours of NRT patches or visually-identical placebo. For safety reasons, women were instructed that if they smoked at all while using patches, the NRT should

be stopped. Participants set a target quit date and were followed up 1 month after this, during hospital admission for delivery or as soon as possible afterwards and at 24 months after delivery.

### Maternal baseline data and smoking behaviour measures
Baseline data included maternal date of birth, age on leaving full-time education, number of cigarettes smoked per day prior to and during pregnancy, Body Mass Index (BMI), the Heaviness of Smoking Index (HSI) - a measure of nicotine dependence which is a six-point scale derived from the time to first cigarette after waking and the number of cigarettes smoked daily.[16] Other data included were previous use of NRT in the current pregnancy and partners' smoking status.

Self-reported smoking data were obtained 1 month after the target quit date and at delivery. Self-reported smoking abstinence at 1 month after the quit date was validated by exhaled CO measurements below 8 ppm while abstinence at delivery was validated by CO measurements and saliva cotinine concentrations below 10 ng/mL. Carbon monoxide is exhaled from the breath after smoking cigarettes and was used to confirm abstinence within the preceding 24 hours. Salivary cotinine on the other hand was used to determine smoking exposure within the last 7 days.[17] No data on tobacco smoke exposure were collected between the 1 month follow-up date and delivery. Self-reported data on the use of nicotine patches were collected 1 month after the target quit date and at delivery; adherence to NRT have been reported elsewhere.[14]

### Infant outcome measures
Birth status (live or stillbirth), gestational age at birth, birth weight, singleton or multiple birth, sex and gestational age at birth were recorded at delivery and have been reported elsewhere.[14] The presence of infant developmental impairment at 24 months was determined using questionnaire responses from participants or from healthcare professionals when there was no participant response; full details have been described previously.[15] Participants reported on their infants' development as assessed within the five domains of the Ages and Stages Questionnaire, Third Edition (ASQ-3): communication, gross motor, fine motor, problem solving and personal-social skills.[15] Participant questionnaires (PQ2) were posted out at 24 months and if no response was received, health professional ones were dispatched to non-respondents' family physicians. Health professional questionnaire (HPQ) items were consistent with domains in participant questionnaires and were designed to be completed with reference to medical or health visitor records. Health professionals completing these questionnaires required relatively little knowledge of the patient. However if they were unable to complete the questionnaire, they were asked to forward these to children's health visitors.[18] Scores from the five ASQ-3 domains and responses to 'non-domain' ASQ-3 items in the participant

questionnaires were used with established norms to categorise infants as having 'developmental impairment' or 'no developmental impairment'. Infants with only HPQ responses were considered to have no developmental impairment when the responses from all questions indicated no potential developmental issues.[15] As shown in the main study, 88% of study participants returned a PQ2 or HPQ. The proportion of participants assessed using the PQ2 or HPQ were similar in the treatment arm and placebo arm. Approximately two-thirds of participants completed and returned the PQ2, while the HPQ was the source of data for children of the remaining participants.

## Measures of tobacco smoke exposure

As outlined above, smoking behaviour measures were collected at randomisation, 1 month after a target quit date and at the end of pregnancy. From these data, we derived the following categories to represent different levels of tobacco smoke exposure during pregnancy: (i) Number of cigarettes smoked daily during pregnancy, at randomisation, (ii) HSI at randomisation, categorised as high (4 to 6) or low (0 to 3), (iii) Women who reported not smoking between their target quit date and 1 month, validated using exhaled CO and (iv) Women who reported not smoking between their target quit date and the end of pregnancy or shortly afterwards, validated by either or both of exhaled CO and saliva cotinine at the end of pregnancy and also by exhaled CO at 1 month after their quit date.

As all trial participants smoked at the time of randomisation, exposure category (ii) was intended to dichotomise those who smoked more heavily and less heavily in the first weeks of pregnancy prior to study randomisation. For categories (iii) and (iv), we hypothesised that those demonstrating validated abstinence at both 1 month and the end of pregnancy (category iv) would have a lower overall tobacco smoke exposure than those for whom abstinence was only validated at 1 month after their quit date. However, it should be noted that no biochemical test can validate smoking cessation for a prolonged period and some women in either group may have smoked for a time between baseline/study enrolment and 1 month or the end of pregnancy.

## Analyses

Maternal and infant characteristics were analysed by study arm, for participants who had singleton live births and 2 year data on infant developmental impairment. Summary statistics were represented as number (%), mean (SD) and median (IQR) for categorical, normal continuous and non-normal continuous variables, respectively. Univariable logistic regression investigated the association between 2 year developmental outcomes and different measures of maternal tobacco smoke exposure such as baseline number of cigarettes smoked daily in pregnancy, the heaviness of smoking index during pregnancy, maternal smoking abstinence at some point during pregnancy (measured 1 month after the participants'

set quit date) and maternal smoking abstinence during the latter stages of pregnancy (measured at delivery). In multivariable analyses, we adjusted the univariate models for baseline characteristic variables and also controlled for variables that could confound the association between maternal smoking and infant developmental impairment. Potential confounders included in the multivariable analyses were: maternal age, age at which full-time education was completed, maternal BMI, partner's smoking status, infant birth weight, gestational age at birth and study arm (NRT or placebo). Confounder selection was done using the change-in-estimate criteria,[19] and any covariate which changed the effect size of the univariate exposure-outcome model by 10% was considered an important confounder and included in the fully-adjusted model. All analyses were performed using Stata V.14. Significant associations were defined at the p≤0.05 significance level.

## Patient and public involvement

The original SNAP study had a patient and public involvement (PPI) representative who provided PPI insights for the Trial Steering Committee.

## RESULTS

From 1050 pregnant smokers enrolled in the SNAP trial, there were 12 women with twin pregnancies, 14 participants who had foetal deaths, four lost to follow-up and 10 who withdrew consent. Of 1010 women with singleton live births, data on developmental impairment was available for 884 infants. Analyses therefore included maternal and 2 year records of these 884 infants (figure 1). The self-reported median (IQR) number of daily cigarettes smoked by women prior to pregnancy was 19 (10 to 60). At the time of recruitment into the study, the reported median (IQR) number of cigarettes smoked daily in pregnancy was 15 (10 to 20). Of the infants born to participants in the study, 52% were males and less than 10% had low birth weight. Developmental impairment was present in 12.7% of these children (n=112).

Study participants and singleton infants with 2 year outcome data had similar baseline characteristics to participants reported in the full trial cohort.[15] Aside from the fact that participants who were lost to follow-up had higher mean index of multiple deprivation scores implying that they lived in more deprived areas than participants who had data up to 2 years, there were only minor differences in the baseline characteristics between the participants with 2 year data and those who were lost to follow-up.[15]

## Maternal and infant characteristics

The baseline characteristics of the women and children included in the analyses were found to be similar in participants randomised to the NRT or placebo study arm (presented in table 1). The majority of participants included in the study (72.4%) were less than 30 years old; the median BMI (IQR) at randomisation into the study

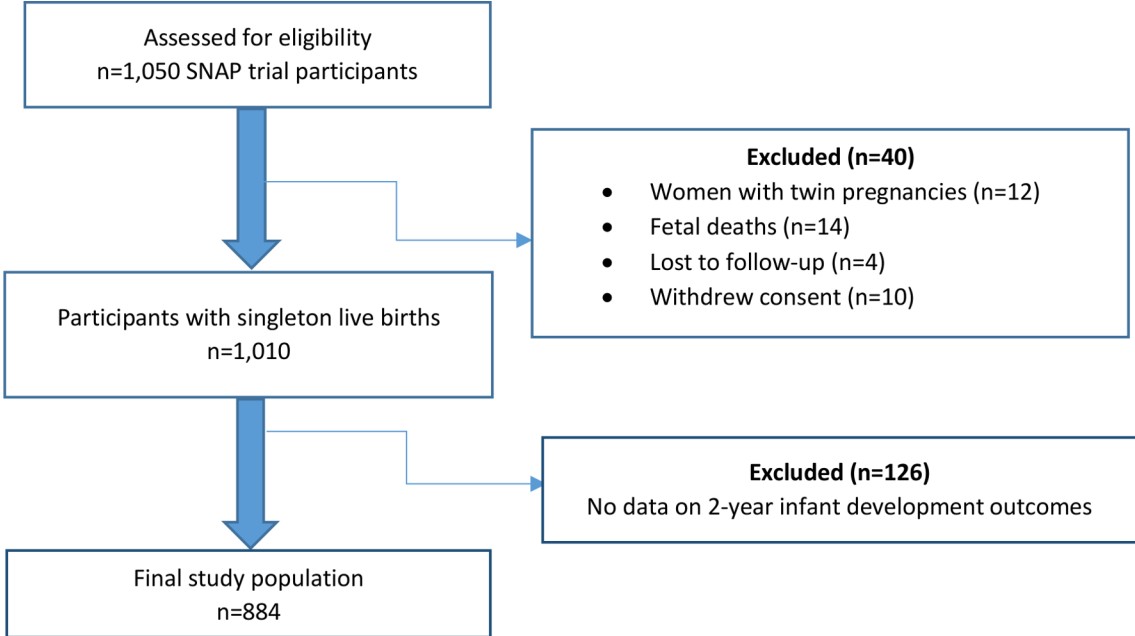

**Figure 1** Flow diagram of study participants. Participants with twin pregnancies (n=12), foetal deaths (n=14), loss to follow-up (n=4) and those who withdrew consent (n=10) were excluded from the study analyses. Among participants with singleton live births, there were no data on 2 year infant outcomes for 126 participants, so these participants were also excluded from the analyses. SNAP, smoking, nicotine and pregnancy.

was 25.9 (22.2 to 30.7). The average age of leaving full-time education was 16.3 years. We used the 'Heaviness of Smoking Index' as a measure of the level of maternal nicotine dependence and 65% of women in the study had low levels of nicotine dependence prior to being recruited in the study. More than two-thirds of the study participants had partners who smoked (67.9%). Overall, 18% of women were validated as abstinent from smoking at some point during pregnancy (measured 1 month after their target quit date) and 8.9% remained abstinent at the latter stages of pregnancy until delivery. There was a statistically significant difference in smoking cessation outcomes between participants in the NRT and placebo arms at the 1 month follow-up but not at delivery.

Analyses of the associations between different measures of maternal tobacco smoke exposure and infant developmental impairment found no statistically significant relationship between maternal smoking exposure and infant developmental impairment at age 2 (table 2). The ORs for developmental impairment in infants of mothers with validated smoking cessation at some point during pregnancy (measured at the 1 month follow-up) and in infants of mothers with validation cessation at delivery, were 0.99 (95% CI 0.59 to 1.66) and 1.55 (95% CI 0.84 to 2.87), respectively. Adjustment for maternal and infant baseline characteristics and covariates did not significantly alter the association between maternal smoking and infant development. Table 2 shows the unadjusted ORs with the different measures of cigarette smoke exposure, as well as the ORs adjusted by maternal and infant baseline characteristics.

## DISCUSSION
### Summary of principal findings
We found no associations between maternal smoking status at different points during pregnancy and infants' developmental impairment at 2 years. Consequently, there is no evidence to support the hypothesis that the better infant development observed within infants born to women who were randomised to NRT in the SNAP trial resulted from smoking cessation induced by nicotine patch use.

### Strengths and limitations
Data for this study are from the largest randomised controlled trial of NRT in pregnancy in which participants, healthcare and research staff were masked to treatment allocation.[14] Follow-up and outcome ascertainment rates were high and there were only minor differences in characteristics of participants lost to follow-up and those remaining in the study at 2 years.[15] It remains the only trial of a smoking cessation intervention in pregnancy to monitor infant outcomes; as such it is the only source of data which could be used to investigate the study hypothesis.

The ASQ-3 is a well-validated screening tool which has a sensitivity of 100% and specificity of 93% for detecting severe developmental delay at 24 months.[20] While some study infants may have been falsely identified as developmentally impaired, there is no reason to believe that such ASQ-3 false-positives would have occurred more frequently in infants of mothers who smoked in pregnancy. While developmental impairment was assessed in some infants using the participant questionnaire, others

**Table 1** Maternal and infant characteristics, by study arm (n=884)

| | Unit | NRT arm (n=443) | Placebo arm (n=441) | P value |
|---|---|---|---|---|
| **Maternal characteristics** | | | | |
| Maternal age (years) at pregnancy | | | | |
| <20 | | 72 (16.25) | 64 (14.51) | |
| 20–24 | | 142 (32.05) | 150 (34.01) | |
| 25–29 | n (%) | 110 (24.83) | 102 (23.13) | 0.275 |
| 30–34 | | 64 (14.45) | 84 (19.05) | |
| 35–39 | | 44 (9.93) | 30 (6.80) | |
| >40 | | 11 (2.48) | 11 (2.49) | |
| Maternal BMI (kg/m$^2$) (n=847) | Median (IQR) | 25.71 (22.2–30.7) | 26.32 (22.5–30.8) | 0.042 |
| Index of multiple deprivation score | Mean (SD) | 32.11 (16.83) | 32.29 (16.84) | 0.754 |
| Maternal age (years) of leaving full-time education (n=872) | Mean (SD) | 16.17 (1.36) | 16.32 (1.70) | 0.95 |
| Partner's smoking status | | | | |
| Non-smoker | | 108 (24.38) | 106 (24.04) | 0.962 |
| Smoker | n (%) | 301 (67.95) | 299 (67.80) | |
| No data on partner's smoking status | | 34 (7.67) | 36 (8.16) | |
| Heaviness of smoking index at study randomisation | n (%) | | | |
| Low index (0–3) | | 284 (64.11) | 292 (66.21) | 0.511 |
| High index (4–6) | | 159 (35.89) | 149 (33.79) | |
| Daily number of cigarettes smoked at study randomisation | Median (IQR) | 13 (10–20) | 15 (10–20) | 0.799 |
| Smoking cessation measured during pregnancy (at 1 month follow-up) | n (%) | | | |
| Did not cease smoking | | 341 (76.98) | 384 (87.07) | <0.0001 |
| Ceased smoking | | 102 (23.02) | 57 (12.93) | |
| Maternal smoking cessation measured at time of delivery | n (%) | | | |
| Did not cease smoking | | 397 (89.62) | 408 (92.52) | 0.131 |
| Ceased smoking | | 46 (10.38) | 33 (7.48) | |
| **Infant characteristics** | | | | |
| Infant sex | | | | |
| Female | n (%) | 217 (48.98) | 205 (46.49) | 0.457 |
| Male | | 226 (51.02) | 236 (53.51) | |
| Birth weight | | | | |
| <2.5kg | | 48 (10.84) | 37 (8.39) | |
| 2.5–3.0kg | n (%) | 111 (25.06) | 104 (23.58) | 0.534 |
| 3.0–3.5kg | | 161 (36.34) | 174 (39.46) | |
| >3.5kg | | 123 (27.77) | 126 (28.57) | |
| Gestational age at birth (weeks) | Median (IQR) | 39.5 (2.1) | 39.5 (2.2) | |

BMI, Body Mass Index; NRT, nicotine replacement therapy.

had the health professional questionnaire as the main source of data on impairment. Although this consequently implies a non-standardised assessment of the outcome in all infants, findings from the main study had shown that the proportion of participants assessed with either the PQ2 or the HPQ were similar in the treatment and placebo arms. Also, there were similarities in the baseline characteristics and birth outcomes of participants who

**Table 2** Multivariate models of the association between maternal tobacco smoke exposure in pregnancy and child developmental impairment at 2 years (analyses restricted to 884 mothers with measures of infant 2 year developmental outcome)

| | ORs (95% CI) for developmental impairment at age 2 | |
| --- | --- | --- |
| | Unadjusted | Adjusted* |
| Daily number of cigarettes smoked at study randomisation | 1.01 (0.98 to 1.04) | 1.00 (0.97 to 1.03) |
| Maternal heaviness of smoking index at study randomisation (high vs low) | 1.30 (0.87 to 1.95) | 1.26 (0.84 to 1.90) |
| Maternal smoking cessation in pregnancy (measured 1 month after randomisation) | 0.99 (0.59 to 1.66) | 1.03 (0.61 to 1.75) |
| Maternal abstinence from smoking during latter stages of pregnancy (y/n) | 1.55 (0.84 to 2.87) | 1.53 (0.82 to 2.87) |

*Adjusted for maternal age, gestational age at birth and infant birth weight.

returned the PQ2 and those for whom the HPQs were completed.

Although we adjusted for some potentially confounding factors which are known to be associated with infant development such as maternal socioeconomic status,[21] maternal education,[22] low birth weight,[23] maternal exposure to passive smoking[24] and maternal obesity,[25] we were unable to adjust for those on which we had no data, such as maternal nutrition,[26] depressive illness,[27] stress and anxiety[28] and alcohol consumption[29] which could also potentially influence infant development but which were not measured in the SNAP trial. There remains therefore, the potential for residual confounding.

We undertook secondary analyses of randomised controlled trial data and the original sample size was intended to have sufficient power only for detecting differences in validated smoking cessation rates at the end of pregnancy. Analyses presented here may not have sufficient power to demonstrate associations between smoking and infant development; however, one would expect to find one or more such associations if, the relationship between NRT use and infant development which was demonstrated within the same data, were principally explained by smoking cessation caused by NRT.

There may have been some misattribution of smoking status. Measures of smoking status used were biochemically-validated at two different time points in pregnancy. Validation with carbon monoxide at the earlier time point could only eliminate smoking within the previous 24 hours[30]; and at delivery, with saliva could only do so for the previous 7 days.[17] Consequently, some participants may have relapsed or smoked occasionally between these times[31]; however we believe it is reasonable to assume that women who were validated as not smoking both at the end of pregnancy and at 1 month after randomisation would have lower overall tobacco smoke exposure than those who were not. As recruitment into the study occurred between 12 and 24 weeks gestation, the timing of smoking status measurements in relation to gestation will have differed between participants but there is no evidence that smoking is more or less harmful at any point in pregnancy. Although these were the best available data

for analyses, the assumptions outlined above need to be considered when interpreting study findings.

As most trial participants continued to smoke, analyses might have been more illuminating if data on the intensity of women's smoking during pregnancy had been available. For example, if exhaled CO measurements or numbers of cigarettes smoked daily in pregnancy had been available for non-abstinent trial participants, these data could have been used to investigate whether smoking intensity in pregnancy was associated infant development. Increasing smoking intensity, reflected by exhaled CO levels, is associated with reduced infant birth weights,[32] so an association with development seems possible.

Our analyses did not incorporate postnatal environmental cigarette smoke exposure which may have an independent influence on infant development.[33] We know that there is continued maturation of the connections between brain regions after birth, with intense brain white matter myelination in the first postnatal months which becomes progressively less rapid through toddlerhood until young adulthood.[34] Additionally, postnatal exposure to environmental tobacco is associated with emotional and conduct behavioural problems in school aged children[35] and so may have an aetiological role in infant developmental impairment. As maternal prenatal and postnatal smoking are strongly correlated, this would increase the likelihood of finding associations between smoking in pregnancy and infant development; as none were found there is no suggestion of a large impact on findings. Nonetheless, there is a need to account for environmental cigarette smoke exposure in future studies.

### Comparison with literature
As this is an original analysis in a unique trial database there is very little literature with which study findings can be compared. Previous observational studies have found prenatal tobacco exposure to be associated with offspring adverse consequences including behavioural[36] and hyperactivity[37 38] problems, language and reading deficits,[37] conduct disorders,[39] newborn basic perceptual skills[40] and intellectual impairment in childhood[41] and later life.[42] However, other studies which adjusted for

potentially confounding factors such as maternal education[43 44] and parental socioeconomic status[45] have not. Consistent with these latter studies, our findings found no evidence that the improved developmental outcomes in infants of pregnant smokers who used NRT in the SNAP trial[15] were explained by smoking cessation. As preclinical studies show that nicotine is neurotoxic and can adversely affect the developing central nervous system,[11 12] it seems unlikely that these better outcomes are due to a direct protective effect of NRT on the developing foetus. Also, compliance rates in the NRT patch and placebo study arms were low in the SNAP trial (7.2% and 2.8% compliance, respectively, at 1 month). Perhaps, it is possible that nicotine used in pregnancy has no impact on infants' developmental outcomes and the apparently protective effect of NRT on developmental outcomes observed in the SNAP trial occurred by chance.

## CONCLUSION

We found no evidence that the better 2 year developmental outcomes in offspring of pregnant women randomised to NRT patch use in the SNAP trial was due to the smoking cessation caused by NRT. Future research which takes account of women's intensity of smoking in pregnancy as well as important maternal confounders and postnatal exposures, is needed to investigate how NRT use in pregnancy may exert this effect on infant development.

**Acknowledgements** We gratefully acknowledge the SNAP trial team members. In addition to listed authors, the complete trial team includes: Investigators: Kim Watts, Jim Thornton, Jo Leonardi-Bee, John Britton, Paul Aveyard, Michael Coughtrie, Christine Godfrey, Clare Mannion and Neil Marlow. Research staff: Janet Brown, Yvette Davis, Anne Dickinson, Caroline Dixon, Fiona Holloway, Joanne Lakin, Jayne Platts, Farzana Rashid, Amanda Redford, Cara Taylor. Principal investigators (in recruiting centres): Jonathan Allsop, Simon Cunningham, Karen Glass, Vince Hall, Khaled Ismail, Margaret Ramsay. Midwife leads in recruiting centres): Sheena Appleby. Denise Bailey, Linda Gustard, Emma Haworth, Grace Hopps, Amanda Lindley, Chris Kettle, Colleen Pearce, Dymphna Sexton-Bradshaw, Julia Savage, Sandra Smith, Sheila Taylor, Alison Witham. Primary Care Trust & NHS Stop Smoking Services' Staff: Barbara Brady, Michelle Battlemuch, Wendy Dudley, Rochelle Edwards, Lorraine Frith, Indu Hari, Catriona Holden, Linda Hoskyns, Paul Jackson, Giri Rajaratnam, Deborah Richardson, Lucy Wade, Maureen Whittaker. QMC Pharmacy: Bernie Cook, Sheila Hodgson (Lead Pharmacist), Lisa Humphries, Bernie Sanders (Qualified Person). University of Nottingham Clinical Trials Unit: Dan Simpkins. University of Dundee: Sheila Sharp.

**Contributors** BI had full access to all of the data in the study and takes responsibility for the integrity of the data and the accuracy of the analyses. BI also wrote the first draft of the manuscript. LRV contributed to the initial stages of data organisation, study concept and design and proofread and approved the full manuscript prior to submission. TC contributed to the study hypothesis and design, data interpretation and analyses, assisted with writing up the 'introduction' section and proofread and approved the final manuscript prior to submission. JT, SL and SC contributed to the data interpretation and analyses of this study as well as revision of the manuscript prior to submission.

**Funding** This trial was supported by a grant from the NIHR HTA Programme (project number 06/07/01). BI's clinical academic lectureship is fully funded by the NIHR. TC, LV, SC and, SL and TC are members of the UK Centre for Tobacco and Alcohol Studies (UKCTAS). TC, LV, SC and JT are members of the NIHR School for Primary Care Research. Professor Coleman is a National Institute for Health Research Senior Investigator.

**Disclaimer** Views and opinions expressed are those of the authors and do not necessarily reflect those of the National Institute for Health Research (NIHR) Health Technology Assessment (HTA) Programme, the NIHR, the National Health Service (NHS) or the English Department of Health.

**Competing interests** None declared.

**Patient consent for publication** Not required.

**Ethics approval** National research ethical approval for the SNAP trial was granted by Oxfordshire Research Ethics Committee A, and additional local approvals for each recruitment centre and Clinical Trial Authorisation (CTA) approval was from the MHRA (CTA number: 03057/0002/001-0001). All participants gave written informed consent, including access to medical records for maintaining contact and for the follow-up of their child's health status.

**Provenance and peer review** Not commissioned; externally peer reviewed.

**Data sharing statement** This study analysed data from the smoking and nicotine in pregnancy (SNAP) trial. All requests for this data should be directed to the snap trial team members. Trial participants did not give consent for data sharing but the data are anonymised and risk of identification is low.

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
