## [Reviewer comments · BMJ Open]

ARTICLE DETAILS

TITLE (PROVISIONAL)	Is the apparently protective effect of maternal nicotine replacement therapy (NRT) used in pregnancy on infant development explained by smoking cessation?: secondary analyses of a randomised controlled trial
AUTHORS	Iyen, Barbara; Vaz, Luis; Taggar, Jaspal; Cooper, Sue; Lewis, Sarah; Coleman, Tim

VERSION 1 – REVIEW

REVIEWER	Melissa Suter Baylor College of Medicine USA
REVIEW RETURNED	02-Aug-2018

GENERAL COMMENTS	The authors present a secondary analysis of an important and rigorous RCT, the SNAP trial. The authors address an important question using data from this trial; namely, does abstinence from smoking for a period during pregnancy correlate with the finding from their previous publication revealing lack of developmental delay in the NRT cohort. A major concern upon review of this manuscript is that their principle findings from their 2012 paper focused on subjects who received NRT, and not the placebo. However, in this analysis, the authors do not take into account who received NRT vs those who only had placebo. Another concern in both the abstract and the first figure is that only 884 subjects had data at 24 months of age, and the authors report that this RCT had 1050 subjects. It needs to be stated clearly that this analysis only included a subset of the original cohort. The authors collected data on maternal smoking at the 24 month follow-up, but did not include postnatal smoke exposure in their analysis. This would be a powerful addition to the manuscript. They addressed that they did not perform the analysis in the discussion section under limitations, but I am unsure why the analysis wasn't conducted. The authors found a significant effect of infant sex on developmental delay, which bears the need for reporting, but the results are curious. The authors addressed this sufficiently in the discussion. A minor suggestion would be to stratify the period of smoking
--

	cessation by weeks gestation to see if there is a specific developmental window that could account for the findings observed in their 2012 study.
--	---

REVIEWER	Lucinda England Centers for Disease Control and Prevention, USA
REVIEW RETURNED	08-Aug-2018

GENERAL COMMENTS	General comments This is a well-written paper on a relevant topic. Because these findings are so closely linked to those of the previously published paper, more details on the earlier findings are needed. For example, in their interpretation of the findings, the authors discussed the potential effects of nicotine vs. cessation on developmental outcomes. However, the manuscript does not include information on compliance with the intervention, which would be very helpful. The original trial results note that only 7% of women assigned to NRT and 3% of those assigned to placebo used medications for more than one month. How many patches did women use? I found the suggestion in the introduction and discussion that nicotine might be neuroprotective to be curious. This is probably the most unlikely scenario, and I wonder if it is even worth putting forward, especially with such limited data on exposure and no biological data from animal studies to back it up. I suggest removing mention of that possibility and focusing on the more likely scenario, such as that the earlier findings were spurious, or women in the treatment arm reduced their overall exposure even though cessation rates at the end of pregnancy. Specific comments Abstract Background: Can you add the main results that lead to this paper—that women randomized to NRT had higher early quit rates compared with those randomized to placebo, and but no difference at the end of pregnancy? The abstract would make more sense with this information. Methods: please add a few words about how developmental outcomes were assessed. Results: define or rephrase “heaviness of smoking.” Conclusions—I think it would be more accurate to state that there was no evidence to support that NRT treatment improved outcomes through smoking cessation. The authors haven’t ruled out that it improved outcomes through reduction in exposure. Introduction Page 4: What was the nature of the dose-response relationship between NRT adherence and impaired development—how was adherence measured or defined? Did it include abstinence from smoking while on NRT? Where does the 8% come from? Can the authors use the OR and
---

95% CI instead, which would be less ambiguous and easier to find in the earlier paper?

The authors state it is unlikely that the association between NRT and development is due to nicotine having beneficial effects on the fetus. I agree with the statement, but there are other more likely potential explanations in addition to that and benefits from quitting. Women on NRT (and their fetuses) would have less exposure to products of combustion, for example. I wonder why the authors even mentioned the possibility that protective effects of nicotine might explain the findings—I don't think that idea would occur to most people and perhaps it adds unnecessary confusion.

Methods

Page 8: It isn't clear to my why the authors didn't adjust for treatment arm in their analysis. Could they explain?

How many children had a health professional outcome measure vs. parents' assessment? How comparable are the two measures with respect to sensitivity and specificity or did the age of assessment differ between the two groups? Did the source of the assessment differ by treatment arm or cessation status? This information is in earlier papers, but it would be helpful to summarize it in the current paper.

The authors referred to looking at "developmental outcomes" but appear to only report "developmental impairment." Did the authors consider looking at individual domains of development in addition to an overall outcome? How was impairment defined with the two measurement tools?

Page 7: please define heaviness of smoking index (what do 0-3 and 4-6 refer to?), and be clear about what "baseline number of cigarettes smoked per day during pregnancy" refers to (an average of the number of cigarettes smoked per day at study recruitment?). Make certain this information matches what is in the table in terms of how things are labeled and what definitions are used. For example, is "baseline number of cigarettes smoked per day during pregnancy" the same as "daily number of cigarettes smoked per day during pregnancy"?

Is there any way the investigators could also look at reduction in cigarette use rather than just cessation? This could be an explanation for an association with NRT. In an earlier study of NRT and smoking cessation, the authors found that although cessation rates didn't change, the treatment arm showed improvements in birth weight, possibly because of reduced exposure to products of combustion. Wisborg K, Henriksen TB, Jespersen LB, Secher NJ. Nicotine patches for pregnant smokers: a randomized controlled study. *Obstet Gynecol.* 2000 Dec;96(6):967-71.

Page 8: I would add gestational age to the list of potential confounders, and z-score for birthweight-for-gestational age. This could help quantify the contributions of gestational age and fetal growth restriction. I would run models with and without variables that could be part of a causal pathway between smoking and developmental outcomes. This is particularly important because fetal growth restriction increases with the number of cigarettes smoked per day, so helps get at the reduction-without-cessation issue.

	Did the authors have access to any additional variables that might be important—multiple birth, or medical, pregnancy, or delivery complications? Page 10: the author collected but did not analyze long-term abstinence, between delivery and developmental assessment. Since exposure to SHS has been associated with some cognitive outcomes, it would be helpful to include this variable as a potential confounder, even if the authors don't have other data on SHS exposure. Discussion Again, the authors could consider looking at change in the amount smoked in NRT vs. placebo groups, not just cessation. The authors stated on page 8 that maternal BMI was included in models. On page 13, they state they had no data on pre-pregnancy obesity. What maternal BMI did they use in models? Page 15: suggest adding additional references for research on SHS and cognitive outcomes, such as Yolton, K., Dietrich, K., Auinger, P., Lanphear, B.P., Hornung, R., 2005. Exposure to environmental tobacco smoke and cognitive abilities among U.S. children and adolescents. Environ. Health Perspect. 113 (1), 98–103. Again, I find it curious to put forward the idea that nicotine exposure could be neuroprotective when there is no other data to support that suggestion and given that there are other explanations for the earlier finding, such as cutting down on smoking (and thus reducing exposure to products of combustion), residual confounding, or chance. I wouldn't even put forward the idea that nicotine is neuroprotective. Everything we know from animal research supports the opposite. Please add the outcome measures as a limitation. Ideally, all children would have undergone developmental assessment under the same conditions. Associations between male sex and a number of developmental disorders have been described. The authors could refer to the following, for example: Jichong Huang, Tingting Zhu, Yi Qu, and Dezhi Mu. Prenatal, Perinatal and Neonatal Risk Factors for Intellectual Disability: A Systemic Review and Meta-Analysis. PLoS One. 2016; 11(4): e0153655. Imac Maria Zambrana, Francisco Pons, Patricia Eadie, Eivind Ystrom. Trajectories of language delay from age 3 to 5: persistence, recovery and late onset. International Journal of Language & Communication Disorders, 2013; DOI: 10.1111/1460-6984.12073
--	--

REVIEWER	John Carlin Murdoch Children's Research Institute & University of Melbourne
REVIEW RETURNED	22-Oct-2018

GENERAL COMMENTS	This paper reports a secondary analysis of a randomised trial of
--

nicotine replacement therapy (NRT) carried out among women smoking in pregnancy (the SNAP trial), and it is directly motivated by results reported previously for 2-year outcomes from that trial (Lancet Respir Med 2014). In the abstract of this earlier paper the main result reported is that “In the NRT group, 323 (73%) of 445 infants had no [developmental] impairment compared with 290 (65%) of 443 infants in the placebo group (odds ratio [OR] 1.40, 95% CI 1.05–1.86, $p=0.023$).” This marginally statistically significant finding is interpreted as if it reflected a true (implicitly causal, given the randomisation) effect, and the current manuscript seeks to identify mechanisms that might explain the effect. Clearly the intended purpose of NRT is to assist women to quit smoking, so it is plausible that those allocated to the active NRT arm of the trial would have quit or reduced smoking to a greater extent than those allocated to the placebo arm, which might then have led to desirable outcomes in the offspring, such as those examined in the secondary analysis reported in the submitted manuscript.

Unfortunately there was very little evidence that women in the NRT arm did in fact achieve substantially lower levels of smoking than those in the placebo arm. In fact, Table 4 of the 2014 paper shows that the proportion of women who reached cessation targets was at most 11% in the NRT arm and only slightly lower in the placebo arm. With so few women actually reducing their smoking, it seems a priori very unlikely that allocation to NRT could cause an effect on any outcome by way of its effect on smoking. Importantly, if there were a prima facie case for such an effect then the appropriate way to investigate it would be by a form of mediation analysis, not by simply ignoring the original randomised allocation and looking for an overall association between the proposed intermediary (in this case, the woman’s smoking) and the outcome. As it stands, by only reporting analyses that ignore the randomised allocation, the current paper does nothing more than many other observational cohort analyses of associations between smoking and/or nicotine exposure in pregnancy and child outcomes. And the lack of evidence of an association may well be due to the limited variation in smoking levels among this cohort of women.

Furthermore, given the results just summarised, the most likely explanation for the apparent difference in developmental impairment between the trial arms at 2 years seems to be chance variation. This is not acknowledged in the 2014 paper where there appears to be a completely false implication that because $p<0.05$ then this must be a “real” effect. (For detailed explanations and references on this unfortunately widespread misconception, see for example Greenland et al, Eur J Epidemiol. 2016; 31:337-350.) As somewhat of an aside, I suggest there may be further doubts surrounding this original “finding” because of the fact that the secondary analysis of the original trial may have looked at many different endpoints, with a selective focus on this one (perhaps the “most significant”?). The registered trial summary available from the ISCRTN registry specifies a wide range of broadly defined outcomes at 2 years(*) and

	does not identify developmental impairment as the "primary outcome at 2 years" as stated in the Lancet Respir Med paper. The possibility that the observed difference in developmental impairment was just a quirk of random variation is acknowledged in the current paper (p.15, lines 45-52), but the logical implication of this is that there may actually be no basis for the (data-derived) "hypothesis" stated at the beginning of the Discussion section (p.13, lines 11-17). Similarly, the final line of the conclusion (p.16) calling for "further research" is not justified, if the perfectly reasonable explanation of chance variation is accepted. In conclusion, although this is a nicely written paper (and I would like to be more generous), I believe it is based on a flawed premise which means its null results are to be expected and are not really informative. Minor point:  - The "unadjusted" results in Table 2 repeat those shown in Table 1, while the "adjusted" associations differ so little that it hardly warrants a whole table to display them. (*) from http://www.isrctn.com/ISRCTN07249128:  4. Early childhood outcomes:  4.1. Behaviour and development at 2 years 4.2. Disability at 2 years 4.3. Respiratory symptoms at 2 years
--	---

VERSION 1 – AUTHOR RESPONSE

Responses to reviewer 1

A major concern upon review of this manuscript is that their principle findings from their 2012 paper focused on subjects who received NRT, and not the placebo. However, in this analysis, the authors do not take into account who received NRT vs those who only had placebo.

Response: We acknowledge that the original analyses reported outcomes separately in subjects who received NRT and placebo. This study however sought to secondarily investigate the unexpected finding of significantly unimpaired development among infants born to women who used NRT. Our hypothesis for the secondary analysis was that the difference in infant development outcomes was due to reduced cigarette smoke exposure in the NRT group. Hence, we investigated the association between maternal tobacco smoke exposure and infant outcomes at 2 years in all study participants with 2-year infant outcomes. In investigating this association, we took into account, baseline maternal tobacco exposure (cigarettes/day), baseline heaviness of smoking, partner's smoking status and maternal abstinence at different stages in pregnancy, irrespective of whether these subjects were randomised to the NRT or control arm in the original study.

Another concern in both the abstract and the first figure is that only 884 subjects had data at 24 months of age, and the authors report that this RCT had 1050 subjects. It needs to be stated clearly that this analysis only included a subset of the original cohort.

Response: A total of 1,050 pregnant smokers enrolled in the SNAP trial. Among 1,010 women with singleton live births, data on developmental impairment was available for 884 infants. The study therefore included the maternal and 2-year outcome records of these 884 infants. Amendments have been made in the abstract, results and first figure to reflect these numbers. As stated in the results section of the manuscript, subjects with 2-year outcome data had similar baseline characteristics with participants reported in the full cohort.

The authors collected data on maternal smoking at the 24 month follow-up, but did not include postnatal smoke exposure in their analysis. This would be a powerful addition to the manuscript. They addressed that they did not perform the analysis in the discussion section under limitations, but I am unsure why the analysis wasn't conducted.

Response: The SNAP study had shown that at 2 years, prolonged smoking abstinence was self-reported in only 3% of mothers in the NRT group and 2% of mothers in the placebo arm. Unlike smoking status in pregnancy (one month after the target quit date) and at delivery, these 2-year smoking outcomes were not validated and no biochemical measures of maternal smoking and no biomarkers of tobacco smoke exposure were obtained from the infants. As a result, no definite conclusions about maternal smoking or infants' exposure to environmental tobacco smoke could be made from the data.

A minor suggestion would be to stratify the period of smoking cessation by weeks' gestation to see if there is a specific developmental window that could account for the findings observed in their 2012 study.

Response: Participants in the SNAP trial were 12 to 24 weeks gestation on enrolment. Following randomisation to either transdermal NRT or visually-identical placebos (which they started to use on their agreed quit date), validated smoking status of all participants were assessed at the same time period - one month after the agreed quit date, and on hospital admission for delivery.

Analyses of the association between maternal gestation in weeks at enrolment and validated one-month smoking cessation rate, found no statistically significant difference in cessation rate among women randomised at different weeks of gestation (p -value for chi-squared test statistic = 0.584). Also, results of stratifying the analyses between maternal one-month cessation and 2-year developmental impairment, by gestational age at randomisation, did not find appreciable difference in the stratum-specific odds ratios for 2-year developmental impairment associated with maternal one-month smoking cessation.

Responses to reviewer 2

Because these findings are so closely linked to those of the previously published paper, more details on the earlier findings are needed. For example, in their interpretation of the findings, the authors discussed the potential effects of nicotine vs. cessation on developmental outcomes. However, the manuscript does not include information on compliance with the intervention, which would be very helpful. The original trial results note that only 7% of women assigned to NRT and 3% of those assigned to placebo used medications for more than one month. How many patches did women use?

Response: As stated in the manuscript, participants in the SNAP trial were randomised to receive up to an 8-week course of either 15mg/16 hours of NRT patches or visually identical placebos. Women who smoked at all while using the patches, were instructed to stop the use of the NRT patch. During follow-up, one month after the agreed quit date, only 7.2% of women assigned to NRT and 2.8% of women assigned to placebo, reported continued use of the trial medication. There was only self-

reported use of NRT patches recorded for a small subset of participants, which limits the usefulness of these data in any analysis. The introduction section of the manuscript has been amended to reflect the compliance rates in both groups of women.

I found the suggestion in the introduction and discussion that nicotine might be neuroprotective to be curious. This is probably the most unlikely scenario, and I wonder if it is even worth putting forward, especially with such limited data on exposure and no biologically data from animal studies to back it up. I suggest removing mention of that possibility and focusing on the more likely scenario, such as that the earlier findings were spurious, or women in the treatment arm reduced their overall exposure even though cessation rates at the end of pregnancy.

Response: Findings in the SNAP study that infants born to women randomised to the NRT group were less likely to have impaired development compared to those born to women in the placebo group, was an unexpected one. Results of the SNAP trial had also shown a significant difference in validated quit rates 1 month after randomisation such that participants in the NRT group had doubled their cessation rates for at least 4 weeks during their second trimester (OR 2.10, 95% CI 1.49-2.97). Our suggestion in the paper was that whereas a neuroprotective effect of nicotine from NRT was highly implausible, we hypothesized that the better outcomes in those infants might be due to smoking cessation caused by NRT.

We strongly agree that it is very unlikely that nicotine might be neuroprotective and so, have amended the sentence in the introduction and discussion so that the message is less misleading

Specific comments (Reviewer 2)

Abstract

Background: Can you add the main results that lead to this paper—that women randomized to NRT had higher early quit rates compared with those randomized to placebo, and but no difference at the end of pregnancy? The abstract would make more sense with this information.

Response: As suggested, the main results from the SNAP trial have now been included in the abstract

Methods: please add a few words about how developmental outcomes were assessed.

Response: The Methods section has been amended to indicate how developmental outcomes were assessed

Results: define or rephrase “heaviness of smoking.”

Response: The heaviness of smoking index (HSI) is a well-known measure of nicotine dependence derived from the time to first cigarette after waking and the number of cigarettes smoked daily. This is defined in the Methods section of the manuscript.

Conclusions: I think it would be more accurate to state that there was no evidence to support that NRT treatment improved outcomes through smoking cessation. The authors haven't ruled out that it improved outcomes through reduction in exposure.

Response: We believe that the suggested statement conveys the same message as the previous statement. The abstract conclusion has however been amended as suggested.

Introduction

Page 4: What was the nature of the dose-response relationship between NRT adherence and impaired development—how was adherence measured or defined? Did it include abstinence from smoking while on NRT?

Response: As described in the main results of the SNAP trial(1), logistic regression analysis was used to explore the dose-response relationship between self-reported adherence with nicotine patches and infants' survival without impairment. Three categories were created for this analysis:

zero adherence (i.e. allocated placebo patch or reported zero nicotine patches used)
For participants who reported use of at least one nicotine patch,

A category representing below the median reported adherence of 10 nicotine patches (i.e. 1–10 days adherence).

A category representing above the median reported adherence of 10 nicotine patches (i.e. 11-56 days adherence)

Results of the analyses showed a dose–response relation between use of NRT patches in pregnancy and infant outcomes at 2 years compared with infants born to participants who did not use nicotine patches. There was no difference in outcomes of infants born to women who reported using between one and 10 NRT patches. However, those infants born to women who reported using between 11 and 56 NRT patches were more likely to have no impairment (OR [adjusted for partner smoking status] 1.72, 95% CI 1.22–2.57, $p=0.004$).

Where does the 8% come from? Can the authors use the OR and 95% CI instead, which would be less ambiguous and easier to find in the earlier paper?

Response: In the main study, infants born to women in the NRT group were significantly more likely to have survived with no impairment than those receiving placebo (OR 1.40, 95% CI 1.05-1.86). The 8% reported in the manuscript was an error and has been amended. We have also reported the findings as odds ratio and 95% confidence intervals.

The authors state it is unlikely that the association between NRT and development is due to nicotine having beneficial effects on the fetus. I agree with the statement, but there are other more likely potential explanations in addition to that and benefits from quitting. Women on NRT (and their fetuses) would have less exposure to products of combustion, for example. I wonder why the authors even mentioned the possibility that protective effects of nicotine might explain the findings—I don't think that idea would occur to most people and perhaps it adds unnecessary confusion.

Response: Please see response to comment (2) above. This section has been amended as suggested

Methods

Page 8: It isn't clear to me why the authors didn't adjust for treatment arm in their analysis. Could they explain?

Response: Please see response to reviewer 1's comment 1

How many children had a health professional outcome measure vs. parents' assessment? How comparable are the two measures with respect to sensitivity and specificity or did the age of assessment differ between the two groups? Did the source of the assessment differ by treatment arm

or cessation status? This information is in earlier papers, but it would be helpful to summarize it in the current paper.

Response: As shown in the main study(1), 88% of study participants returned a PQ2 (participant questionnaire) or HPQ (health professional questionnaire). The proportion of participants assessed with the PQ2 or HPQ were similar in the treatment arm and placebo arm. Approximately two-thirds of participants completed and returned the PQ2, while the HPQ was the source of data for children of the remaining participants. Items in the HPQ (health professional questionnaire) were consistent with PQ2 and the age of assessment using either of these tools was similar. There were only minor differences between the baseline characteristics and birth outcomes of participants who returned PQ2 and those for whom HPQs were completed. This has been summarised in the methods section of the manuscript.

The authors referred to looking at “developmental outcomes” but appear to only report “developmental impairment.”

Response: The objective of the study was to investigate relationships between maternal smoking in pregnancy and infant development. Although we refer to developmental outcomes, the main study outcome of interest in the study was developmental impairment.

Did the authors consider looking at individual domains of development in addition to an overall outcome? How was impairment defined with the two measurement tools?

Response: In the main study, analyses of scores from the individual domains of the ASQ-3 was done, as this showed effect sizes similar size and direction. Details of how the absence, presence or severity of impairment was determined with the two measurement tools, are as stated in the main study and can be found in the study statistical analyses plan(1)

Page 7: please define heaviness of smoking index (what do 0-3 and 4-6 refer to?), and be clear about what “baseline number of cigarettes smoked per day during pregnancy” refers to (an average of the number of cigarettes smoked per day at study recruitment?). Make certain this information matches what is in the table in terms of how things are labeled and what definitions are used. For example, is “baseline number of cigarettes smoked per day during pregnancy” the same as “daily number of cigarettes smoked per day during pregnancy”?

Response: Please see above response to comment in the abstract section above. As stated above and in the methods section of the manuscript, the heaviness of smoking index (HSI) is a well-known measure of nicotine dependence(2). The HSI is a six-point scale which was categorized into 2, to dichotomise those who smoked more heavily (4-6) from those who smoked less heavily in the first few weeks of pregnancy prior to study randomisation.

The term “baseline number of cigarettes smoked” represents the self-reported average number of cigarettes smoked daily per day at recruitment into the study. This term has been amended to “number of cigarettes smoked at randomisation”, for clarity. The labels in the table have also been amended to match these terms.

Is there any way the investigators could also look at reduction in cigarette use rather than just cessation? This could be an explanation for an association with NRT. In an earlier study of NRT and smoking cessation, the authors found that although cessation rates didn't change, the treatment arm showed improvements in birth weight, possibly because of reduced exposure to products of combustion. Wisborg K, Henriksen TB, Jespersen LB, Secher NJ. Nicotine patches for pregnant smokers: a randomized controlled study. *Obstet Gynecol.* 2000 Dec;96(6):967-71.

Response: The participant outcomes of interest in the main study were abstinence from smoking at different time points between the quit date and delivery.

Aside from smoking status/ cigarette smoke exposure at randomisation into the study, the only maternal information collected during follow-up, and which was therefore available for secondary analyses in this study were, self-reported and validated smoking status as well as NRT use, of all participants at one month after the agreed quit date, and on hospital admission for delivery.

Lack of smoking exposure between the 2-time points when data were collected, is a limitation which has been stated in the study manuscript. We agree that reduction in cigarette in women who used NRT, rather than cessation, could partly explain the finding of better 2-year infant developmental outcomes. As stated in the manuscript however, we believe it is reasonable to assume that women who were validated as not smoking both at the end of pregnancy and at one month after randomisation would have lower overall tobacco smoke exposure than those who were not.

Page 8: I would add gestational age to the list of potential confounders, and z-score for birthweight-for-gestational age. This could help quantify the contributions of gestational age and fetal growth restriction. I would run models with and without variables that could be part of a causal pathway between smoking and developmental outcomes. This is particularly important because fetal growth restriction increases with the number of cigarettes smoked per day, so helps get at the reduction-without-cessation issue.

Response: As all participants were between 12 and 24 weeks gestation at randomisation, we acknowledge as a limitation, that timing of smoking measurements would have differed between participants. However, there is no evidence that smoking is more or less harmful at any point in pregnancy.

As stated in response to reviewer 1's comment 4, further sensitivity analyses were done among participants randomised at different gestational age. We found no statistically significant difference in cessation rate among women randomised at different weeks of gestation (p-value for chi-squared test statistic = 0.584).

Also, stratified analyses (stratified by gestational age at enrolment) of the association between maternal one-month cessation and 2-year developmental impairment did not find appreciable difference in the stratum-specific odds ratios for 2-year developmental impairment associated with maternal one-month smoking cessation.

The main study of the SNAP trial(3) had compared birth outcomes between participants randomised to NRT and placebo arms and found that for singleton births, mean birth weight – z score, rates of preterm birth, low birth weight and congenital abnormalities were similar in the two groups. Did the authors have access to any additional variables that might be important—multiple birth, or medical, pregnancy, or delivery complications?

Response: These variables were available in the dataset. However, these outcomes in the NRT and placebo arm, have been reported in previous studies of the SNAP trial(1, 3). Also, participants with multiple births were excluded from the study, so only those with singleton births were included.

Page 10: the author collected but did not analyze long-term abstinence, between delivery and developmental assessment. Since exposure to SHS has been associated with some cognitive outcomes, it would be helpful to include this variable as a potential confounder, even if the authors don't have other data on SHS exposure.

Response: Please see response to Reviewer 1's comment 3

Discussion

Again, the authors could consider looking at change in the amount smoked in NRT vs. placebo groups, not just cessation.

Response: Please see response to this comment in the Method section above

The authors stated on page 8 that maternal BMI was included in models. On page 13, they state they had no data on pre-pregnancy obesity. What maternal BMI did they use in models?

Response: The maternal BMI was measured at randomisation. No measure of BMI or adiposity prior to pregnancy was available for participants

Page 15: suggest adding additional references for research on SHS and cognitive outcomes, such as Yolton, K., Dietrich, K., Auinger, P., Lanphear, B.P., Hornung, R., 2005. Exposure to environmental tobacco smoke and cognitive abilities among U.S. children and adolescents. *Environ. Health Perspect.* 113 (1), 98–103.

Response: Thank you for the suggestion. This reference has been included in the manuscript.

Again, I find it curious to put forward the idea that nicotine exposure could be neuroprotective when there is no other data to support that suggestion and given that there are other explanations for the earlier finding, such as cutting down on smoking (and thus reducing exposure to products of combustion), residual confounding, or chance. I wouldn't even put forward the idea that nicotine is neuroprotective. Everything we know from animal research supports the opposite.

Response: Please see response to comment 2 above

Please add the outcome measures as a limitation. Ideally, all children would have undergone developmental assessment under the same conditions.

Response: This has been included as a limitation in the manuscript

Associations between male sex and a number of developmental disorders have been described. The authors could refer to the following, for example:

Jichong Huang, Tingting Zhu, Yi Qu, and Dezhi Mu. Prenatal, Perinatal and Neonatal Risk Factors for Intellectual Disability: A Systemic Review and Meta-Analysis. *PLoS One.* 2016; 11(4): e0153655.

Imac Maria Zambrana, Francisco Pons, Patricia Eadie, Eivind Ystrom. Trajectories of language delay from age 3 to 5: persistence, recovery and late onset. *International Journal of Language & Communication Disorders*, 2013; DOI: 10.1111/1460-6984.12073

Response: Thank you for the above references on studies which show an association between male sex and developmental impairment. These have been included in the manuscript.

Response to reviewer 3

Thank you for the comments.

With regards to the comment “with so few women actually reducing their smoking, it seems a priori very unlikely that allocation to NRT could cause an effect on any outcome by way of its effect on smoking”, we wish to point out that NRT had a substantial impact on cessation in early pregnancy. At delivery however, there was no significant difference in cessation rate between women who used NRT and those who did not. Our study hypothesis therefore sought to investigate whether the difference in infant development outcomes was due to reduction in cigarette smoke exposure among some of the women in our cohort compared to others.

With regards to the non-adjustment for treatment arm in our analyses, please see response to reviewer 1’s comment 1

References

1. Cooper S, Taggar J, Lewis S, Marlow N, Dickinson A, Whitemore R, et al. Effect of nicotine patches in pregnancy on infant and maternal outcomes at 2 years: follow-up from the randomised, double-blind, placebo-controlled SNAP trial. *Lancet Respir Med.* 2014;2(9):728-37.
2. Heatherton TF, Kozlowski LT, Frecker RC, Rickert W, Robinson J. Measuring the heaviness of smoking: using self-reported time to the first cigarette of the day and number of cigarettes smoked per day. *Br J Addict.* 1989;84(7):791-9.
3. Coleman T, Cooper S, Thornton JG, Grainge MJ, Watts K, Britton J, et al. A Randomized Trial of Nicotine-Replacement Therapy Patches in Pregnancy. *N Engl J Med.* 2012;366(9):808-18.

VERSION 2 – REVIEW

REVIEWER	Melissa Suter Baylor College of Medicine USA
REVIEW RETURNED	14-Jan-2019

GENERAL COMMENTS	The authors have responded thoroughly to all concerns.
--

REVIEWER	Lucinda England Centers for Disease Control and Prevention, USA
REVIEW RETURNED	12-Feb-2019

GENERAL COMMENTS	This is an interesting and important analysis. My main suggestion is that although these data are unique, there are still quite a few shortcomings, including a lack of data on postnatal potential confounders. Therefore, the findings are better treated as hypothesis generating. The authors can help advance the field by noting how some of these shortcomings can be addressed in future studies. Introduction Page 4, line 34: consider adding other substances relevant to pregnant women, such as carbon monoxide.
--

Line 55-60: the “4%” doesn’t align with an OR of 1.40. If there is a percentage point increase of 4%, please add that finding. If there is a typo, please correct.

Page 6, line 6: I assume baseline “date of birth” refers to maternal age at birth? Can this be clarified?

Paragraph starting on line 18: can the authors clarify how often self-report and CO or cotinine were discrepant?

Line 25: why does it say that saliva cotinine was “estimated”? It might be helpful to include the period of time after last smoking that the CO measurement will be positive so the reader has a sense of the potential for false negatives of the testing. This doesn’t appear until the discussion but would be useful to have earlier.

3rd paragraph: What was the age range at which the 24 month child assessments were completed? I assume they weren’t all done on time. If provider administered items were sent to non-responders, then children of non-responders were likely older at the time of administration, correct? How much older?

Line 56: can the authors elaborate on how the HPQ’s were completed? Am I correct in understanding that the family physician was given the tool and completed it based on information in medical records? What if there weren’t any recent visits? It seems like there is the potential for different length of follow up in infants in the two arms.

Page 8

Did the authors include study arm in initial models? If not, why not?

I’m unclear as to why gestational age wasn’t included in adjusted models. Rather than adjusting for birth weight, I suggest adjusting for gestational age and birthweight adjusted for gestational age (you could use z-scores, for example).

Was BMI in models prepregnancy?

Does the list of “potential confounders” (line 43) include those in the final models, or all variables evaluated for inclusion in models? It is unclear what variables were available for analysis—later, IMD is mentioned, but is unclear whether it was included in stepwise models. Table 1 does not appear to include a complete list either—it would be helpful to see complete results for univariable analysis.

Were there any postnatal exposures available for analysis?

Page 10: It is stated that only infant sex was retained in models in addition to “a priori defined variables.” On page 8, variables stated to be included in multivariable analyses are listed, but not described specifically as the “a priori defined variables.” It would help to clarify this.

Discussion

Page 13: the authors state that there is no reason to believe that false positive developmental assessments would have occurred

	more frequently in infants of mothers who continued to smoke during pregnancy. However, if continuing smokers have lower education (this isn't clear from the tables), they could have less accurate reporting. Page 14: the authors state they adjusted for SES but I don't see that in the final variables. They then discuss the inability to adjust for maternal mental health conditions, alcohol consumption, etc., but note that results were not near significant without adjustment, so their inability to adjust for confounders is not likely to explain their findings. Again, it is important not to be too dismissive of the potential role of confounders. Future studies should include adjustment for these factors and important postnatal exposures. Suggest adding references for CO and cotinine (lines 49-50). Page 15: The authors state they were unable to adjust for maternal smoking after pregnancy, but that it is correlated with prenatal smoking so likely not an important confounder here. However, many women quit smoking during pregnancy and relapse after delivery, and fathers smoking status or SHS from other sources could also be important. Therefore, I would not dismiss SHS as an important potential confounder but rather would state that future studies need to account for SHS exposure. Pages 15-16, discussion of explanations for the findings: The discussions of cessation and a potential protective effect of nicotine seem reasonable. The introduction includes a summary of patch and placebo use in the SNAP trial (7.2% and 2.8% for one month, respectively). This information could be re-introduced in the discussion as part of the discussion of whether NRT may have played a role. Page 16: The results of a sensitivity analysis should be included in the results section, unless this is in reference to earlier publications, in which case it should be referenced. I suggest adding to the conclusions the authors' recommendations for future research to address unresolved issues. Suggest changing "apparently protective effect of NRT patch" to "improved outcomes in offspring of women randomized to NRT..." Finally, as it is written now, there is no suggestion of next steps or future directions.
--	---

VERSION 2 – AUTHOR RESPONSE

Reviewer: 1

Please leave your comments for the authors below

The authors have responded thoroughly to all concerns.

Responses to reviewer: 2

Please leave your comments for the authors below

This is an interesting and important analysis. My main suggestion is that although these data are unique, there are still quite a few shortcomings, including a lack of data on postnatal potential confounders. Therefore, the findings are better treated as hypothesis generating. The authors can help advance the field by noting how some of these shortcomings can be addressed in future studies.

Specific comments

Introduction

Page 4, line 34: consider adding other substances relevant to pregnant women, such as carbon monoxide.

Response: This line has been amended and some of the harmful substances in tobacco smoke have been included in this sentence

Line 55-60: the "4%" doesn't align with an OR of 1.40. If there is a percentage point increase of 4%, please add that finding. If there is a typo, please correct.

Response: Thank you for your comment. The 4% was an error and has been amended to 40% in the revised manuscript.

Page 6,

line 6: I assume baseline "date of birth" refers to maternal age at birth? Can this be clarified?

Response: In the section titled "Maternal baseline data and smoking behaviour measures", the date of birth does refer to the maternal date of birth. The term maternal has been included for further clarification.

Paragraph starting on line 18: can the authors clarify how often self-report and CO or cotinine were discrepant?

Response: Discrepancies in smoking validation were dealt with during data analyses of the original SNAP trial study. For example, 7 of the 208 participants who admitted to not smoking in the previous 24 hours before the one-month visit, had exhaled CO readings at or above the threshold of 8ppm. Only participants who self-reported (a) smoking abstinence within the time windows studied (between quit date and 1 month follow-up or between quit-date and delivery), (b) smoking abstinence of at least 24 hours prior to the follow-up date and (c) whose self-reported smoking abstinence were verified by exhaled CO measurements and/or saliva cotinine concentrations, were assumed to have ceased smoking at the defined time periods and included in the original SNAP data analyses. As such, only these SNAP subjects with validated smoking status were included in our secondary analyses.

A limitation which we acknowledged in the manuscript is that biochemical validation with exhaled carbon monoxide at the one-month follow-up could only eliminate smoking within the previous 24 hours. At delivery, validation with saliva cotinine concentrations could eliminate smoking for the previous seven days. Consequently, some participants may have relapsed or smoked occasionally between these time periods without being detected.

Line 25: why does it say that saliva cotinine was "estimated"?

Response: In the SNAP trial, cotinine measurements were taken from participants' saliva samples. The word estimated has been deleted from the manuscript.

It might be helpful to include the period of time after last smoking that the CO measurement will be positive so the reader has a sense of the potential for false negatives of the testing. This doesn't appear until the discussion but would be useful to have earlier.

Response: These have now been included in the revised methods section of the manuscript

3rd paragraph: What was the age range at which the 24 month child assessments were completed? I assume they weren't all done on time. If provider administered items were sent to non-responders, then children of non-responders were likely older at the time of administration, correct? How much older?

Response: The 2-year infant follow-up questionnaires were dispatched to parents four weeks before the infants' 2nd birthday. This was followed by a 2nd birthday card, two weeks later, with reminders for non-respondents. Non-respondents after 2 questionnaires were followed up by telephone call. Health professional questionnaires (HPQ) were only sent out if parents failed to complete the questionnaires (PQ) despite repeated contact. The ASQ-3 24-month questionnaire used at the 2-year follow-up, is valid for use from 23 months 0 days to 25 months 15 days. If used at 24 months, ASQ scores require no adjustment for prematurity. In the SNAP trial, care was taken to ensure that the ASQ-3 questionnaire was only used within infants for whom it would be valid, and only responses obtained during this window were included in the study¹. Furthermore, a comparison of the characteristics between participants who returned the PQs and those for whom the HPQ were completed, found very minor differences between the groups' baseline characteristics or birth outcomes²

Line 56: can the authors elaborate on how the HPQ's were completed? Am I correct in understanding that the family physician was given the tool and completed it based on information in medical records? What if there weren't any recent visits? It seems like there is the potential for different length of follow up in infants in the two arms.

Response: the health professional questionnaires were very short questionnaires sent to participants' general practitioners (GPs), containing only items to measure children's disability which correspond to those on the parent questionnaires. These items were designed to be easily completed using medical or health visitors' records. Health professionals completing these questionnaires required relatively little knowledge of the patient. If GPs were unable to complete the questionnaire, they were asked to forward these to children's' health visitors (HV)¹. As stated in the manuscript, full details of how developmental impairment was determined using questionnaire responses from participants or healthcare professionals have been described in previous work and in the published SNAP study protocol. This information has now been included in the revised manuscript.

Page 8

Did the authors include study arm in initial models? If not, why not?

Response: Study arm was not included in the submitted manuscript but has now been included in this revised version. Inclusion of study arms in analyses however made no difference to the association between maternal smoking exposure and 2-year infant development.

I'm unclear as to why gestational age wasn't included in adjusted models. Rather than adjusting for birth weight, I suggest adjusting for gestational age and birthweight adjusted for gestational age (you could use z-scores, for example).

Response: We have now included gestational age at birth, in addition to birth weight, in adjusted models. The methods section reflects this, and the results of analyses where these were included in the final multivariable model, are shown in Table 2 in the results section.

Was BMI in models prepregnancy?

Response: The maternal body-mass index were measured at randomisation into the study. Researchers in the SNAP trial had no access to pre-pregnancy BMI so the only available measure of BMI was the BMI at time of randomisation into the study (between 12-24 weeks gestation). Though not an ideal measure, we have used it as a proxy measure for adiposity in study participants.

Does the list of “potential confounders” (line 43) include those in the final models, or all variables evaluated for inclusion in models? It is unclear what variables were available for analysis—later, IMD is mentioned, but is unclear whether it was included in stepwise models. Table 1 does not appear to include a complete list either—it would be helpful to see complete results for univariable analysis.

Response: The list of potential confounders in line 43 (Analyses subsection of manuscript methods) were those evaluated for inclusion in the final model. Following the review comments, the study analyses were repeated, with inclusion of additional suggested variables in the list of potential confounders. Table 1 and 2 have been updated accordingly to reflect these changes.

Were there any postnatal exposures available for analysis?

Response: There were no postnatal smoking exposures available for analyses. The SNAP study had shown that at 2 years, prolonged smoking abstinence was self-reported in only 3% of mothers in the NRT group and 2% of mothers in the placebo arm. Unlike smoking status in pregnancy (one month after the target quit date) and at delivery, these 2-year smoking outcomes were not validated and no biochemical measures of maternal smoking and no biomarkers of tobacco smoke exposure were obtained from the infants. As a result, no definite conclusions about postnatal maternal smoking or infants' exposure to environmental tobacco smoke could be made from the data.

Page 10: It is stated that only infant sex was retained in models in addition to “a priori defined variables.” On page 8, variables stated to be included in multivariable analyses are listed, but not described specifically as the “a priori defined variables.” It would help to clarify this.

Response: We apologise for the lack of clarity in the methods and results section of the submitted manuscript. As these analyses have now been repeated, these sections have been amended.

Discussion

Page 13: the authors state that there is no reason to believe that false positive developmental assessments would have occurred more frequently in infants of mothers who continued to smoke during pregnancy. However, if continuing smokers have lower education (this isn't clear from the tables), they could have less accurate reporting.

Response: We agree that participants with lower education may have less accurate reporting of developmental impairment using the ASQ-3 tool. Our analyses however adjusted for maternal age of leaving full-time education as well as index of multiple deprivation and neither of these altered the association between maternal smoking and 2-year infant development.

Page 14: the authors state they adjusted for SES but I don't see that in the final variables. They then discuss the inability to adjust for maternal mental health conditions, alcohol consumption, etc., but note that results were not near significant without adjustment, so their inability to adjust for confounders is not likely to explain their findings. Again, it is important not to be too dismissive of the potential role of confounders. Future studies should include adjustment for these factors and important postnatal exposures.

Response: We adjusted for socio-economic status using the index of multiple deprivation but this did not significantly alter the association between maternal smoking and 2-year developmental impairment. It was therefore not included in the final multivariable model. As none of the baseline or potential confounding variables altered this association, we decided to include only the baseline maternal and infant characteristics in the final adjusted model.

We acknowledge the potential for residual confounding and agree that future studies should include more confounders and important postnatal exposures. As such, we have removed the comment which states that "... no analyses had borderline significant results so such confounding would need to have substantial impact to mask associations".

Suggest adding references for CO and cotinine (lines 49-50).

Response: References have been added as suggested

Page 15: The authors state they were unable to adjust for maternal smoking after pregnancy, but that it is correlated with prenatal smoking so likely not an important confounder here. However, many women quit smoking during pregnancy and relapse after delivery, and fathers smoking status or SHS from other sources could also be important. Therefore, I would not dismiss SHS as an important potential confounder but rather would state that future studies need to account for SHS exposure.

Response: We agree that maternal smoking after pregnancy as well as other environmental cigarette smoke exposure are important confounders which would future studies need to account for. Although the SNAP trial had very low quit rates, we agree too that many women who quit smoking during pregnancy, relapse after delivery. This paragraph has been amended with a sentence stating the need for future studies to account for SHS exposure.

Pages 15-16, discussion of explanations for the findings: The discussions of cessation and a potential protective effect of nicotine seem reasonable. The introduction includes a summary of patch and placebo use in the SNAP trial (7.2% and 2.8% for one month, respectively). This information could be re-introduced in the discussion as part of the discussion of whether NRT may have played a role.

Response: Information about patch and placebo compliance in the SNAP trial has now been included in this section to further buttress the argument that better outcomes among infants in the NRT arm are not likely due to an effect of NRT use.

Page 16: The results of a sensitivity analysis should be included in the results section, unless this is in reference to earlier publications, in which case it should be referenced.

Response: The sensitivity analyses reported in the previously submitted version was done to further assess the association between gender and developmental impairment. Following a revision of the study analyses with confounder selection using the change in estimate criteria, infant gender did not meet the criteria for inclusion as a confounder. This section has therefore been removed from the discussion.

I suggest adding to the conclusions the authors' recommendations for future research to address unresolved issues. Suggest changing "apparently protective effect of NRT patch" to "improved outcomes in offspring of women randomized to NRT..." Finally, as it is written now, there is no suggestion of next steps or future directions.

Response: Thank you for the suggestion. The manuscript conclusion has been rewritten with the inclusion of recommendations for further research.

References

1. Protocol for the SNAP (Smoking Nicotine and Pregnancy) Trial Final version 70 2009.
2. Cooper S, Taggar J, Lewis S, et al. Effect of nicotine patches in pregnancy on infant and maternal outcomes at 2 years: follow-up from the randomised, double-blind, placebo-controlled SNAP trial. *Lancet Respir Med* 2014;2(9):728-37. doi: 10.1016/S2213-2600(14)70157-2